# Informed consent and trial prioritization for clinical studies during the COVID-19 pandemic. Stakeholder experiences and viewpoints

**Stefanie Weigold, Susanne Gabriele Schorr, Alice Faust, Lena Woydack, Daniel Strech** [ID] *

QUEST Center for Responsible Research, Berlin Institute of Health (BIH) at Charité – Universitätsmedizin Berlin, Berlin, Germany

* daniel.strech@bih-charite.de

**Data Availability Statement:** Full transcripts cannot be shared publicly because of participants privacy. Excerpts of the transcripts relevant to the study are provided within the paper (see

## Abstract

### Background

Very little is known about the practice-oriented challenges and potential response strategies for effective and efficient translation of informed consent and study prioritization in times of a pandemic. This stakeholder interview study aimed to identify the full spectrum of challenges and potential response strategies for informed consent and study prioritization in a pandemic setting.

### Methods

We performed semi-structured interviews with German stakeholders involved in clinical research during the COVID-19 pandemic. We continued sampling and thematic text analysis of interview transcripts until thematic saturation of challenges and potential response strategies was reached.

### Results

We conducted 21 interviews with investigators, oversight bodies, funders and research support units. For the first topic informed consent we identified three main themes: consent challenges, impact of consent challenges on clinical research, and potential strategies for consent challenges. For the second topic prioritization of clinical studies, we identified two main themes: perceived benefit of prioritization and potential strategies for prioritization. All main themes are further specified with subthemes. A supplementary table provides original quotes from the interviews for all subthemes.

### Discussion

Potential response strategies for challenges with informed consent and study prioritization partly share common ground. High quality procedures for study prioritization, for example, seem to be a core response strategy in dealing with informed consent challenges. Especially in a research environment with particularly high uncertainty regarding potential

supplementary table S2). The informed consent process assured interviewees that only de-identified quotes from the interview would be shared. As the full interview transcripts, even with de-identification of names or institutions, contain potentially identifying or sensitive interviewee information or information about the interviewee's institution, we cannot upload the transcripts. The study and its data protection policy have been approved by the Research Ethics Committee and the Data Protection Office of the Charité. Data requests can be sent to the Charité Research Ethics Committee, which approved this study (approval number EA4/006/21).

**Funding:** DS received funding for this project from the Federal Ministry of Education and Research (BMBF, www.bmbf.de). Grant number:01KI20123. The funder had no role in study design, data collection and analysis, decision to publish, or preparation of the manuscript.

**Competing interests:** The authors have declared that no competing interests exist.

treatment effects and further limitations for valid informed consent should the selection of clinical trials be very well justified from a scientific, medical, and ethics viewpoint.

## Introduction

One of the most challenging parts of a pandemic such as COVID-19 is the lack of evidence-based knowledge about the prevention, diagnosis, and treatment of this new disease. In a very short time, therefore, a worldwide need for research is emerging. The COVID-19 pandemic resulted in thousands of clinical trials on the same disease being planned, reviewed, funded, conducted and published worldwide within a few months [1]. As of June 2020, more than 2300 trials were registered in the clinicaltrials.gov registry, and more than half of these were interventional trials. This extremely high number of clinical trials, combined with the intense time pressure, very uncertain evidence, and pronounced infection control measures, equally led to an amplification of many research ethics challenges and, in some cases, even pandemic- or COVID-19-specific challenges [2]. From a legal and ethical perspective, it is undisputed that even in pandemic or other exceptional situations, research ethical standards must be adhered to in order to protect study participants [3].

Based on our own survey results [2], a scoping literature review and informal discussion with key informants for clinical research on COVID-19 in Germany two topics that are of particular practical and ethical relevance were identified: challenges with informed consent and challenges with prioritizing clinical studies.

The informed consent process shall allow eligible patients to make their own judgments about benefits, risks/burdens, and other relevant aspects that come with study participation. Based on these judgements the patients make informed decisions about whether to participate or not. Valid informed consent exists only if the patients concerned were a) competent to make decisions in the situation in question, b) received the relevant information, c) understood it, and d) were free from coercion to decide for or against consent [3, 4]. Critical care settings in general bear several challenges for ensuring a valid consent process due to difficult patient- and context-related circumstances that decrease decision making capacity [5, 6]. A pandemic situation with social distancing measures in place and high uncertainty regarding potentially effective treatments can further aggravate these challenges. From the beginning of the COVID-19 pandemic until July 2021, Germany experienced several waves of infections with daily new cases reaching over 30,000 in winter 2020. In response, the German government implemented strict lockdown measures, including closures of non-essential businesses, schools, and public spaces. As the vaccination campaign progressed in 2021, the number of daily new cases began to decline, and restrictions were gradually eased. By July 2021, the country was in a phase of reopening.

At the same time the efficient and effective recruitment of eligible patients for clinical studies to investigate pandemic-specific prevention or therapeutic approaches is of utmost importance. Prioritization of clinical studies that is the ranking of more or less important trials might become relevant, for example, due to a shortage of eligible patients or available investigators and trial nurses for conducting trials. If all planned trials recruit at the same time but few patients are eligible, clinical trials might face recruitment failures or trialists compete in recruitment activities. A recent review of registered trials showed that many clinical trials, for example in Germany, failed due to insufficient recruitment [7]. Prioritization might also be important if too many studies are conducted in an inefficient or even harmful way. An explorative review or registered trials showed that many trials investigated the same therapeutic approach at the same time [8].

While some conceptual papers discussed COVID-19 specific challenges for informed consent [9] and study prioritization [10], very little is known about how these challenges and response strategies for effective and efficient translation of informed consent and study prioritization have played out in practice. The objective of this stakeholder interview study, therefore, was to identify the spectrum of challenges and potential response strategies for informed consent and study prioritization in a pandemic setting. The results of the interview study shall inform the development of practice-oriented guidance for informed consent and trial prioritization as a component of pandemic preparedness activities. Respective national activities are, for example, the German PREPARED (Prepardness and Pandemic Response in Deutschland) but also trials-focused network activities such as the German NUKLEUS platform for developing standards to plan, conduct and evaluate large-scale clinical and epidemiological studies nationwide.

## Methods

### Ethical approval

All steps of analysis in this study involving human participants were in accordance with the ethical standards of the research ethics committee of Charité Berlin (Approval No: EA4/006/21) and with the 1964 Helsinki Declaration and its later amendments.

### Participants

We aimed to interview key stakeholders involved in clinical research during the COVID-19 pandemic until thematic saturation is reached. Key stakeholders include clinical investigators, clinicians from clinical trial centers who deal with COVID-19 patients on a daily basis (both intensive care and non-intensive care wards including internal medicine), clinic directors, patient representatives as well as members of research ethics committees, regulatory bodies, public sponsors, guardianship courts and authorities. The recruitment of interviewees followed a purposive sampling approach. We identified representatives of the above mentioned stakeholder groups via the network of the QUEST Center, the AKEK (German working group of research ethics committees), and the PRECOPE advisory board that included representatives from umbrella organizations for clinical research, health care, and patient organizations. We further identified potential interview participants via snowballing using recommendations from interviewees.

### Data collection

All potential interview participants received an email including study information and the informed consent document. All participants provided written informed consent. We conducted the interviews via videoconferencing with the platform MS Teams between March 2021 and July 2021. Interviews were conducted in German and lasted between 30 and 60 minutes. One team member led the interview (LW, SW or DS, all trained in qualitative research methods), while another team member observed and made notes on its process and content (LW, SW, DS or AF). The research team, comprised of individuals with expertise in medicine, public health, and bioethics. The team did not have personal acquaintances with the interview partners. The decision to include an observer during interviews was made to foster reflective discussions on the interview responses. All interviews were guided using a semi-structured topic guide (S1 Table). It focused on challenges and strategies for informed consent and prioritization in the context of Covid-19 trials and was pilot tested with researchers of our institution. After each interview, the team conducted a peer-debriefing. All interviews were video

**Table 1. Demographic data of interview participants.**

| Variable | | Frequency (n) |
|---|---|---|
| Sex | Female | 7 |
| | Male | 14 |
| Age | 30–34 | 3 |
| | 35–44 | 7 |
| | 45–50 | 3 |
| | 50+ | 8 |
| Function | Principal investigators (PI) of Covid-19 clinical study | 8 |
| | Non-PI investigators | 1 |
| | Physicians (at university hospitals) | 7 |
| | Medical University-based clinical research coordinator | 1 |
| | Members of research ethics committees | 3 |
| | Member of public funding body | 1 |

recorded and saved as an audio file. A transcription company transcribed the audio file under a confidentiality agreement. MAXQDA 2020 was used to process and manage the interviews and enter coding results.

We conducted 21 semi-structured in-depth interviews with 21 participants to reach thematic saturation. Altogether, we contacted and invited 51 stakeholders. Our sample included physicians working in clinical research (principal investigators and non-principal investigators), heads of clinical departments that were responsible for the enrollment of patients with COVID-19 in clinical studies, representatives from clinical study centers, members of research ethics committees, and a member of a public funding body. For complete demographics see Table 1.

## Data analysis

We employed qualitative content analysis according to Mayring (2010) [11]. This method involves categorizing the interview content into predefined categories, ensuring reliability and validity through a controlled process, and then interpreting the results in the context of the research question. The analysis was characterized by an inductive development of themes. The transcripts were initially coded by SW. To ensure accuracy and consistency, AF verified the coding on the first 10 transcripts. Coding commenced immediately upon receipt of the first transcript. Notes from observers helped to interpret the transcripts and were available for the whole research team. This iterative coding approach informed our purposive sampling strategy for subsequent interview participants and helped us ascertain when we had achieved thematic saturation. LW provided oversight and actively participated in the early stages of the coding process. All codes and discrepancies within the coding were discussed and further modified in an iterative process in several group meetings as well as written feedback rounds within the research team (SW, LW, AF, SGS, DS) until consensus was reached. The entire analysis was conducted in German. For the purpose of this paper, relevant quotes were translated into English.

For the first topic informed consent we identified three main themes: consent challenges, impact of consent challenges on clinical research, and potential strategies for consent challenges. For the second topic prioritization of clinical studies, we identified two main themes: perceived benefit of prioritization, potential strategies for prioritization. All main themes were further specified with first and second level subthemes (see Table 2 for an overview).

**Table 2. Qualitative spectrum of topics regarding the informed consent for or prioritization of clinical studies on Covid-19.**

| Informed Consent<br>Main themes and first-level subthemes | Second-level subthemes | Code |
|---|---|---|
| Consent challenges | | |
| Time pressure | Rapid deterioration of the patient's condition | C1 |
| | Reduction of contact time | C2 |
| Isolation, physical distancing, and hygiene conditions | Difficulty in building the doctor-patient relationship | C3 |
| | Need to reduce the number of patient contact | C4 |
| | Restricted accessibility to legal guardians due to quarantine regulations | C5 |
| | Contact restrictions for relatives | C6 |
| Overburdening the patient | Distress due to parallel/competitive study recruitments | C7 |
| | Difficulties in understanding due to complexity | C8 |
| | Restrictions of the patients due to severe symptomatology | C9 |
| | Increased risk of therapeutic misunderstanding | C10 |
| Impact of consent challenges on clinical research | | |
| Heterogenous ethics review | | C11 |
| Delay | | C12 |
| Recruitment failure | | C13 |
| Bias | Excluding critically ill patients introduces important bias | C14 |
| Potential strategies for consent challenges | | |
| Opt-in consent as ethical standard in interventional studies | Relevance for interventional studies | C15 |
| | Conflicts of interest | C16 |
| Consent models for secondary use of patient data | Broad consent model | C17 |
| | Opt-out model | C18 |
| | No consent model | C19 |
| Requirements for patients unable to give consent | Need of consistent IRB requirements | C20 |
| | Deferred and legal proxy consent | C21 |
| | Alternative proxy decision makers | C22 |
| **Prioritization Aspects**<br>Main themes and first-level subthemes | Second-level subthemes | Code |
| Perceived benefit of prioritization | | |
| Lack of explicit prioritization | | P10 |
| Ensuring patient protection | Prioritization could improve risk-benefit assessment for clinical trials | P1 |
| | Prioritization could prevent overburdening of patients | P2 |
| Improving validity and efficiency | Prioritization could reduce the high number of parallel/redundant studies | P3 |
| | Prioritization could improve quality of studies | P4 |
| | Prioritization could prevent trials suffering of insufficient resources and expertise | P5 |
| | Prioritization could reduce inefficiencies due to competitive recruitment | P6 |
| Increasing relevance | Prioritization could support the relevance of clinical studies | P7 |
| | Prioritization could identify patient-oriented knowledge gaps | P8 |
| | Prioritization should include the allocation of biological samples | P9 |
| Potential strategies for prioritization | | |
| Central coordination and collaboration | National board | P11 |
| | National cooperation | P12 |
| | Central registry | P13 |
| | Consultation by Ad-hoc expert groups | P14 |
| | Efficient study planning | P15 |
| | Effective recruitment of special patient groups | P16 |
| | Management of conflict of interest | P17 |

*(Continued)*

**Table 2.** (Continued)

| Study design | Multicentre study design | P18 |
|---|---|---|
| | Adaptive study designs | P19 |
| Alignment with funding | Funding for priority studies | P20 |
| | Calls for priority topics | P21 |
| | Efficient funding procedures | P22 |
| | Fair funding procedures | P23 |
| | Sunstainable funding | P24 |
| Pandemic policy on use/ownership of patient data | Agreement for secondary data use | P25 |
| | Regulation on data use and access | P26 |
| Professionalisation of clinical research | Competence in clinical research | P27 |
| | Fair play among scientists | P28 |

## Results

In the following we give a narrative overview of the identified main and subthemes together with quotes for selected subthemes. The corresponding original quotes from the interviews for all subthemes can be found in S2 Table.

### Informed consent

**Consent challenges.** *Time pressure*. Various interviewees emphasized that they needed to obtain the informed consent under increased time pressure. Often this was due to the "rapid deterioration of the health condition of patients" with COVID-19 (C1).

> Quote (interview 14): *"If you are the only person who can enroll a patient in a trial today, you have to do it today, because tomorrow the patient may not be able to give consent, or it may be too late"*

Furthermore, interviewees indicated the need to "reduce contact time" to minimize the risk of infection (C2).

*Isolation, physical distancing, and hygiene conditions*. Various interviewees indicated "difficulties in building the doctor-patient-relationship" due to the intended contact restrictions and the protective clothing (C3). Within the strict isolation conditions, the "number of patient contacts" and the duration of these were reduced in order to minimize the spread of the virus (C4). In addition, it was highlighted that the "reduced accessibility to legal guardians" (C5) and the "contact restrictions for relatives" (C6) had negative impact on informed consent procedures.

*Overburdening the patient*. Interviewees reported several aspects of the pandemic situation that carried a risk to overburden the patient and thus weaken the quality of the informed consent procedure. Some patients were confronted with "parallel/competitive study recruitment" which could cause distress at the patient side (C7).

> Quote (interview 6): *"For example, doctor A goes to the patient and asks: "Would you like to take part in the XY drug trial? The patient asks for time to think about it. During the reflection period, Doctor B comes and clarifies for the next study. And the patient thinks it's the same trial and says yes. These things can happen."*

Patients often had difficulties in "understanding due to complexity" of informed consent documents (C8). This was further complicated by the pandemic context.

Quote (interview 7): *"Many were also sceptical (about participating in a study for a foreign disease such as COVID-19) [. . .] they were often in a very bad condition and then overwhelmed by all the information you have to give them quickly"*.

The informed consent procedure could especially overburden those patients that already suffered "severe symptomatology" and rapid disease progression (C9). Furthermore, some interviewees mentioned an "increased risk of therapeutic misunderstanding" when the disease is rather novel (C10).

**Impact of consent challenges on clinical research.** Several interviewees reported that the challenges around appropriate consent procedures for COVID-19 trials had an impact on conducting clinical research.

*Heterogenous ethics review*. Some interviewees mentioned "heterogenous ethics review" (C11). One example given by interviewees was an IRB decision that required the exclusion from the trial of those participants who had initially consented but lost capacity during the trial due to disease progression. Other IRBs allowed these participants to remain in the trial.

*Delay and recruitment failure*. Interviewees pointed to the potential "delay" (C12) of conducting trials or even "stops/recruitment failures" (C13) that result, for example, from difficulties to contact legal proxies during a pandemic.

*Bias*. Interviewees further pointed to the risk that even in conducted studies the exclusion of critically ill patients due to consent limitations would "introduce important bias" (C14).

Quote (interview 16): *"[. . .] if someone is unable to give consent at the beginning of the trial: For these patients, I think you have to do everything you can to include them in the trials, because otherwise you get an enormous bias if you leave out the seriously ill"*

**Potential strategies for consent challenges.** *Opt-in consent as ethical standard in interventional studies*. Despite the challenges experienced, several interviewees emphasized the informed consent and the self-determination of patients as the ethical standard with particular "relevance for interventional studies" (C15). In addition to patient-centered considerations, some interviewees addressed the protective function of the informed consent for potential "conflicts of interest" of clinician investigators (C16).

Quote (interview 3): *"Never [change consent in times of crisis]! Doctors always have conflicts of interest."*

*Consent models for secondary use of patient data*. Regarding non-interventional studies, various interviewees suggested the implementation of a "broad consent model" for the use of routine data and biological samples (C17). The opportunity to learn immediately from routine care with legitimate access to routine data was emphasized. By pointing to the potentially high social value of secondary use of patient data during a pandemic interviewees suggested an "opt-out model" (C18).

Quote (interview 14): *"[. . .] such an opt-out solution, I would not find it so strange for register studies"*

Likewise, a "no consent model" for the secondary use of patient data was discussed (P19).

Quote (interview 11): *"And I have to say, for this part, my impression at Covid-19 was that the interest for the general public is so great that it would be helpful to define at a high level what is generally possible for the general public and does not require consent"*

*Requirements for patients unable to give consent.* When discussing how to better prepare for potential future pandemics nearly all interviewees expressed the need to implement strategies for those patients who are unable to give consent. Interviewees highlighted the "need of consistent IRB requirements" especially for study participants who lose their consciousness after prior consent (C20). Several interviewees described the "deferred and legal proxy consent" (C21) as an appropriate option. However they referred how difficult and challenging it is for relatives to make such a proxy decision on trial participation especially in a pandemic situation.

Quote (interview 1): *"Well, I would not necessarily require deferred consent, but a combination of deferred consent and a consultant might make it better, yes; that someone from a different field actually has to look at it. I think in observational studies it's relatively uncritical and in interventional studies I think it's also appropriate"*

Some interviewees mentioned the option of "alternative proxy decision maker" (C22) such as physicians or ethics committees.

## Prioritization aspects

**Perceived benefits of prioritization.** *Lack of explicit prioritization.* Our study did not aim to gather quantitative information about how often our interview participants experienced explicit prioritization on which trials to conduct and which not. We want to highlight, however, that from all 21 interviewees engaged in conducting or coordinating clinical research all except one reported that they did not experience any type of explicit prioritization of clinical studies. Several interviewees stressing this "lack of prioritization" at the same time highlighted the different needs for prioritization as presented above (P10).

Quote (interview 5): *"You're offered something, you don't have anything, and then you just take it, and sometimes there's even overlap. So it's problematic: We have multiple trials that are looking at the same patient population, and that's a problem"*

*Ensuring patient protection.* Interviewees described that if prioritization of clinical trials could help reduce unnecessary risks that might come with participation in less well justified trials this would support patient protection and thus "improve risk-benefit assessment" of planned clinical trials (P1).

Quote (interview 11): *"[. . .] And then came the reconvalescent plasma trial. It is still running. And the impression is: [. . .] Not only are we not making a 'benefit', but we are prolonging the intensive care stay. We are rather creating problems. And finally, piece by piece, the studies came here that showed: "It doesn't help there. It doesn't help there. [..] When do you stop such a study? (Interview 11)"*

Prioritization could also "prevent the overburdening of patients" that are asked by different physicians at the same time to participate in their clinical trials (P2).

*Improving validity and efficiency*. Interviewees mentioned the need to "reduce the number of parallel/redundant studies" that were conducted during the COVID-19 pandemic (P3). Several interviewees further highlighted that the designs of COVID-19 trials often lacked measures to increase the robustness and that the reporting quality of study protocols was low. A more explicit prioritization process might therefore "improve the quality of clinical trials" (P4) and could "avoid that trials are conducted with insufficient resources and expertise" (P5). As mentioned above, in some occasions one patient was asked to participate in several studies illustrating the benefit of prioritization measures to "reduce inefficiencies" due to competitive recruitment (P6).

*Increasing relevance*. Numerous interviewees pointed out that several research questions addressed in COVID-19 studies lacked relevance. Prioritization could therefore "support the relevance" of clinical studies (P7).

Quote (interview 2): *"[. . .] this is indeed a problem and I would have liked the national side to have coordinated this and said that these and the studies have priority because they really take us forward and not the Xth study, which examines something but does not really bring a solution in terms of the pandemic"*

Some interviewees commented on important criteria for prioritization and most of them mentioned clinical relevance as a top criterion. Some interviewees criticized that most studies investigated different drug treatments and only few procedures for COVID-19 such as lung ventilation. Therefore, more explicit prioritization could not only select among already planned studies but also "identify patient-oriented knowledge gaps" (P8). Interviewees mentioned that prioritization also becomes relevant in "allocating the scarce resource of biological samples" such as lung tissue from patients with COVID-19 (P9).

**Potential strategies for prioritization.**   *Central coordination and collaboration*. Interviewees repeatedly emphasized that central coordination via, for example, a "national board" (P11) and "national cooperation" (P12) could be helpful to prioritize clinical studies.

Quote (interview 11): *"[. . .] it would have been really helpful, as people have said in the past, if a committee had been set up very quickly and had had time to really get to know each other. Maybe even nationwide and would have said*: *"These are really the best candidates. These are the ones we're counting on".*

The RECOVERY trial [12] and other WHO studies were mentioned as having created relevant evidence via respective coordination and collaboration efforts. In order to profit from the specific expertise and study ideas at individual university hospitals interviewees proposed a "central registry" (P13). Another interviewee suggested the development of a procedure for "ad-hoc expert groups" (P14). This ad hoc expert groups should be set up for specific urgent topics with members from different institutes that jointly discuss and design a specific study. Interviewees highlighted that central coordination could also improve "efficiency in study planning" (P15) on a national level that might allow Germany to better participate in international activities such as the WHO studies. Furthermore, central coordination was also mentioned as a way to support "effective recruitment of special patient-groups" (P16). Additionally, interviewees highlighted that the "management of conflict of interests" could be supported by more central coordination and collaboration (P17).

*Adapting study design*. Interviewees mentioned that study designs like "multicenter study" (P18) could be helpful to improve recruitment and efficiency:

Quote (interview 16): *"There are three studies running in parallel [. . .]. And of course it would be helpful to combine them, because the number of cases and the range of people included is getting larger. The background of the participants also depends on whether I am in a practice at the university hospital or in a peripheral hospital, but we have not been able to bring the participants together."*

Others mentioned that "adaptive study designs" (P19) could further address the need of better coordination in line with the need for prioritization: "then they just need a fixed established therapy group, such as ACTT [Adaptive COVID-19 Treatment Trial], where a joint design is then developed under fixed coordination, but there must then also be a team behind it" (Interview 18).

*Alignment with funding.* While discussing the issue of prioritization, different interviewees mentioned a lack of public calls for therapeutic studies. Prioritization should align with opportunities for "funding priority studies" (P20) and "calls for priority topics" (P21).

Quote (interview 4): *"There is no funding because there is no pharma. There is no funding because there is no virology, no basis in it, but it is a purely clinical issue. I would like to see a central committee. That's where the questions would come from. Then you say these are the 'most important' questions for us that need to be answered"*

Effective prioritization is further dependent on "efficient funding procedures" (P22) and "fair funding procedures" (P23). The fairness of funding was also contextualized as a consequence of more explicit prioritization if the relevance of certain research questions receives more priority than for example the size of a university. "Sustainable funding" (P24) becomes according to an interviewee important to make the prioritization procedures effective.

*Pandemic policy on use/ownership of patient data.* Several interviewees pointed out that a strong "agreement for secondary use of data and biological samples" is needed for an effective and fair prioritization in the secondary use of patient data and biological samples (P25). To support this agreement a "regulation on data use and access" is one important aspect that should be clarified upfront in preparing for pandemics (P26).

Quote (interview 11): *"Who owns the data? Am I giving away my data? Am I still considered enough there? So in an instant everybody gets the feeling: "I'm giving something away. Am I giving away my rights?"*

*Professionalization of clinical research.* In elaborating strategies and success conditions for prioritization of clinical studies several interviewees pointed out that more "competence in clinical research" (P27) is needed in Germany. During the pandemic, sometimes people with insufficient training in study design and other clinical research related competencies organized and conducted clinical research.

Quote (interview 16): *"Maybe that's part of the problem, that people are now doing clinical research who haven't really done clinical research in the past, and then the collaborations don't work"*

Moreover, interviewees pointed out that the need for "fair play among scientists" (P28) serve as a condition of success for collaborative and effective research that qualifies for prioritization.

## Discussion

This interview study investigated stakeholders' experiences and viewpoints regarding informed consent and trial prioritization for clinical research during the COVID-19 pandemic.

### Informed consent

For the topic of informed consent the interviewees mentioned significant practical limitations that can negatively impact on the conditions for a valid consent. The capacity to make autonomous decisions and the understanding of the relevant information can be impacted by practical limitations such as "time pressure" or "isolation conditions". Further challenges such as competitive recruitment might "overburden patients" and thus impact on competence and understanding. The often high uncertainty about expected benefits and harms of investigated treatments (over 99% of investigated COVID-19 treatments failed [13]) also impact on the "relevant information" condition for a valid consent. The more the validity conditions for informed consent are challenged the less does the consent procedure serves its role of patient protection. Our qualitative study cannot provide an answer to the question of how often obtained consent in COVID-19 studies was invalid. The plausibility of having strongly limited conditions for valid consent during a pandemic urgency situation, however, signals the need to define and implement measures that either facilitate better validity conditions or compensate for the decreased protective function of informed consent. We come back to these compensating measures below.

The challenges around informed consent cannot only affect the protection of trial participants but can also limit the effective and efficient conduct of clinical research and thus affect the society at large. Relevant factors potentially limiting clinical research are recruitment challenges (via delay or failure of studies), biased sampling of less severely affected patients (that have higher chances to give consent in a timely fashion). Here again, our qualitative and thus explorative study can only raise awareness about these challenges but we cannot specify how often specific trials were strongly delayed or failed due to consent challenges. Data on recruitment failures in Germany point to relatively low recruitment rates and a lot of early stopping of trials [7]. The challenges of informed consent could be one contributing determinant beside others. To prepare for future pandemics with maybe even more challenging conditions for valid informed consent procedures further ethical, legal and social discussions including all relevant stakeholder groups should address the question of what modification of consent procedures are acceptable in pandemic emergency situations that strongly require clinical research especially with severely affected patients that are not able to give consent.

Our interview results suggest that secondary use of patient data under a broad consent (opt-in) model or even an opt-out model might resolve some of the above mentioned challenges. A broad consent (opt-in) model that requires additional information and consenting procedures with every COVID-19 patient, adds a further layer of complexity to the already resource constrained health care in a pandemic situation. An exemplary discussion for Low and Middle Income Countries (LMIC) can be found at Singh et al. (2021), who evaluate how conducting biobank research under a waiver of informed consent during public health emergencies is ethically permissible [14]. Germany established a national broad consent model in 2020 [15] and applied also this to a national cohort study of COVID-19 patients. Further research is needed to evaluate how efficient and feasible such broad consent (opt-in) model is.

While secondary use of health data can create additional information, it cannot replace clinical trials. In contrast to Germany, in other countries alternative *consent strategies* were considered legitimate also for clinical trials under the special circumstances of the pandemic.

Repeatedly, interviewees referred to the knowledge gained rapidly by the RECOVERY Study conducted in the UK [12]. To be emphasized is the informed consent agreement, already anchored in the study protocol, which made deferred consent solutions possible by a clinician in consultation with the research ethics committees. Patients who lack capacity to consent due to severe disease (e.g. need ventilation), and for whom a legally designated representative is not available, randomization and consequent treatment will proceed with consent provided by a clinician independent of the clinician seeking to enroll the patient who will act as the legally designated representative. Consent will then be obtained from the patient in case of recovery or the patient"s personal legally designated representative at the earliest opportunity [12]. Van der Graaf et al. discuss the opportunities and challenges of such a deferred consent model and highlight that this model should only replace prospective consent by individual patients if it is cleary demonstrated that the number of competent patients to enroll prospectively is insufficient due to for example a quick disease progression or other reasons [9].

## Prioritization aspects

When reflecting on the urgent need to generate knowledge and at the same time the need to select the most promising studies and avoid wasteful redundancies and inefficiencies in clinical trials, interviewees emphasized the potential value of well-informed prioritization decisions. The common task would be to decide which (often collaborative) trials to launch and to assure that they are robust and create patient-oriented output. A successful, frequently cited example is the already mentioned centrally planned multicenter RECOVERY study [12]. Organized and planned by Oxford University, it was initiated with the National Health System. It recruited 10,000 patients within two months and ultimately produced substantial evidence for the treatment of COVID-19 (ibid.) [16].

With regard to the German context, interviewees saw the German Network University Medicine [17] which was founded during the pandemic to foster cooperation between UMCs, as a fruitful starting point. The network aims to bring together and evaluate action plans, diagnostic and treatment strategies from as many German university hospitals as possible. This bundling of competencies and resources is intended to create structures and processes in the hospitals that ensure the best possible care for COVID-19 patients. In addition, interviewees discussed that non-university institutions and networks established through the Robert Koch Institute, for example, should also be considered as collaboration partners. Within all these institutions, databases and registers have been created in preparation for evidence synthesis in future pandemics, which are helpful for potential decision-making (e.g. CEOsys [18], NAP-KON [19], CODEX [20, 21], STAKOB [22]).

A differentiated approach for possible prioritization criteria was also developed by Meyer et al. [10] who have drafted guidance for research institutions on how ethical consolidation and prioritization of COVID-19 clinical trials might proceed in a three-stage assessment. It first considers whether a study meets the criteria of social value, scientific validity, feasibility, and collaboration. Second, if all threshold criteria are met, the institutional capacity to conduct a study should be examined and assessed. Third, study-specific prioritization criteria should be evaluated, concerning the safety and effectiveness of interventions, the robustness of designs as well as institutional resource utilization, expertise and experiences in clinical practice. Meyer at al. suggest further to check several diversity criteria, which query e.g., several patient life stage and risk factors, a wide range of disease stages, and ensure inclusions from different geographic and sociodemographic regions. For implementing these criteria, "COVID-19 research prioritization process should occur prior to IRB, grants office, and other institutional reviews to avoid wasting those important resources" [10]. In addition, communication of

potential prioritization decisions as well emergency risk communication should be considered [23].

Prioritization of specific studies can not only be helpful for scientists by improving quality but also protect patients by increasing the chance that primarily robust and clinically highly relevant studies are initiated. Prioritization and collaboration efforts to improve the scientific justification of trials conducted in the pandemic context would thus function as an important compensation for the above mentioned challenges with the informed consent procedure.

## Limitations of the study

Our study has the following limitations. Firstly, the content focus on informed consent and study prioritization was set by us. As mentioned in the introduction, our focus on these two topics was justified by literature-based preliminary work, a survey study across German research ethics committees, and consecutive expert consultation. Other research ethics issues such as transparency of research results were partly addressed by other activities from our group [24] but the need for a better qualitative understanding of SWOT was judged by external experts and our group to be highest for the two issues we focused on. Secondly, while we were able to include diverse perspectives from trialists, health care professionals, regulatory bodies and despite our contact with patient organizations we were not able to identify patient representatives that could comment more generally on experiences with informed consent and study prioritization in the pandemic context. We believe that both a qualitative interview study and a quantitative survey of trial participants and/or their legal representatives would be an important follow-up work on our explorative qualitative expert interviews.

## Conclusion

This interview study identified a broad spectrum of challenges and potential response strategies for *informed consent* and *study prioritization in a* pandemic setting. A reason for why our interview results do not contain core differences in stakeholder feedback might be that our interviews did not address details for how to implement response strategies. We believe that positionality and differences across stakeholder feedback would become much more prominent for an interviews study focusing on such implementation questions. Interestingly, the potential response strategies for both informed consent and prioritization share some common ground. Procedures for study prioritization, for example, seem to be a core response strategy in dealing with informed consent challenges. Especially in a research environment with particularly high uncertainty regarding potential treatment effects and further limitations for valid informed consent should the selection of clinical trials be very well justified from a scientific and practical relevance viewpoint. The results of the interview study shall inform the development of practice-oriented guidance for informed consent and trial prioritization as a component of pandemic preparedness. As mentioned above, further patient and stakeholder engagement would be key in the early identification of different stakeholder perspectives on most suitable response strategies.

## Supporting information

**S1 Table. Interview topic guide.**
(PDF)

**S2 Table. Part "Informed consent": Qualitative spectrum of topics with relevance for the informed consent for clinical studies during the Covid-19 pandemic.**
(PDF)

## Acknowledgments

We would like to thank the participating interviewees for their time and willingness to talk to us. We would like to thank further members from our research group for comments on the manuscript prior to submission.

## Author Contributions

**Conceptualization:** Daniel Strech.

**Data curation:** Stefanie Weigold, Lena Woydack.

**Formal analysis:** Stefanie Weigold, Susanne Gabriele Schorr, Alice Faust, Lena Woydack.

**Investigation:** Stefanie Weigold, Alice Faust, Lena Woydack.

**Project administration:** Stefanie Weigold, Susanne Gabriele Schorr, Lena Woydack.

**Supervision:** Daniel Strech.

**Validation:** Daniel Strech.

**Writing – original draft:** Stefanie Weigold, Lena Woydack, Daniel Strech.

**Writing – review & editing:** Susanne Gabriele Schorr, Alice Faust.

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
