## [Decision Letter · Decision Letter 0]

24 May 2023

PONE-D-23-01691Informed consent and trial prioritization for human subject research during the COVID-19 pandemic. Stakeholder experiences and viewpoints.PLOS ONE

Dear Dr. Strech,

Thank you for submitting your manuscript to PLOS ONE. After careful consideration, we feel that it has merit but does not fully meet PLOS ONE’s publication criteria as it currently stands. Therefore, we invite you to submit a revised version of the manuscript that addresses the points raised during the review process.

As you will see, the reviewers have provided a very thorough review of your manuscript, with a number of important suggestions to strengthen your work. I encourage you to consider the suggestion of reviewer 2 to focus the manuscript more on the informed consent angle, and to potentially discuss prioritisation as a theme within that focus. If you opt to keep the paper capturing both major subjects, please provide a strong justification for this pairing. 

We look forward to receiving your revised manuscript.

Kind regards,

Rafael Van den Bergh

Academic Editor

PLOS ONE

Journal Requirements:

Reviewers' comments:

Reviewer's Responses to Questions

**Comments to the Author**

1. Is the manuscript technically sound, and do the data support the conclusions?

Reviewer #1: Yes

Reviewer #2: Yes

2. Has the statistical analysis been performed appropriately and rigorously? 

Reviewer #1: N/A

Reviewer #2: N/A

3. Have the authors made all data underlying the findings in their manuscript fully available?

Reviewer #1: No

Reviewer #2: No

4. Is the manuscript presented in an intelligible fashion and written in standard English?

Reviewer #1: Yes

Reviewer #2: No

5. Review Comments to the Author

Reviewer #1: The review is included in the attachment as it exceeds the 20000 Character limit. The review is included in the attachment as it exceeds the 20000 Character limit. The review is included in the attachment as it exceeds the 20000 Character limit.

Reviewer #2: Thank you for the opportunity to review this manuscript. I enjoyed reading it and the issue addressed - informed consent – is one faced by all researchers, and one which becomes more complex in clinical trials taking place in a rapidly shifting pandemic. The response strategies provided by the authors are practical, feasible and drawn from interview data collected from experts with direct experience in relevant fields.

My overall feeling, however, is that there is the potential for much deeper reflection - I was left wanting more detail from the results. I would like the authors to situate their findings within a wider explanation of why informed consent is important for potential research participants as well as institutions and researchers. The authors note the lack of participant voices as a limitation of the study, but I would still like to see more discussion of the informed consent process as a protective mechanism as well as one which advances scientific knowledge. This would help to balance out the arguments and frame the findings in a more patient-centred way, placing the protection of vulnerable patients (or their data) at the centre and stressing the importance of a thorough and ethical informed consent process around them to improve future health interventions/treatment.

I have two main comments before addressing other minor issues below:

1. I feel that the authors are actually addressing two different issues in one: informed consent and trial prioritisation. This may not be in line with other reviewers, but I wonder if it would be beneficial to divide this into two separate manuscripts or if not, focus on informed consent with prioritisation discussed as one theme/solution. This would give the authors more space to present their data in detail and make it a stronger qualitative piece of work.

2. The way the qualitative data is presented means the substance and detail of it is hidden: quotes and examples are lost and over-summarised. The manuscript would benefit from the results being integrated into the main text. I recognise that word count may be an issue, but most of the sections in the results section are very short and need to be expanded to provide more detail and context.

Introduction/abstract

Could the authors provide a definition or explanation of ‘practice-oriented challenges’?

Whilst I recognise that ‘human subject’ is a common phrase in clinical trial literature, it removes some of the agency of the research participants by reducing them to ‘subjects’. Would it be possible to reword the title or offer an explanation/recognition that the use of the word ‘subject’ can be complicated? Language such as being ‘incapable’ of consenting should also be amended.

Can you provide more epidemiological background on the Covid context in Germany at the time of the study? A few sentences here would help the reader to situate the findings and understand the context in which the key informants were working.

I would also have liked to see more examples of why informed consent and ethics are important in clinical trials, for example, through citing ethics guidelines or describing what the informed consent process means for individual research participants, not just the researchers using their data.

Methods

The methods are well described and appropriate for a study of this kind, which relies on key informant interviews of a technical nature.

Could the authors be consistent in how the study is described? I would be tempted to call it a ‘qualitative research study’ which draws on data from key informant interviews, rather than a qualitative interview study.

Thank you to the authors for being honest about the response rate – this is typical of this kind of research, in which the interviewees are stakeholders in busy academic, coordination and research positions. I would move the response rate into the methods section, and link it to the description of your sampling process.

Would the sub-heading ‘procedure’ be better described as ‘data collection’?

What language were the interviews conducted in, and was any translation required at the same time as transcription (eg German to English)?

The ethical-legal analysis mentioned in the introductory comments of the interview guide is not specifically mentioned in the methods – was this part of the study design? If so, could you mention it in the methods?

Results section

I recognise that the results section is a presentation of the data which is discussed and later reflected upon in the discussion, but it is very short and would benefit from the inclusion of more examples.

Upon reading the manuscript for the first time I assumed that C3, C4 etc were individual interviewee codes, used to prevent key informants from being identified, rather than referring to a theme linked to specific quotes. I would like to see the quotations integrated into the main body of the paper, not only in the table - the table is lengthy and my concern is that readers will not go back and forth to search for the examples, meaning that the impact of the qualitative nature of the data is lost.

I suggest restructuring the results so that each theme is introduced in more detail (eg it jumps from a heading of informed consent, to headings on consent challenges and time pressure without any description of the content to follow). I would like to see more ‘linking’ sentences such as ‘many of the interviewees described the time pressures they faced in obtaining consent…’ or ‘this was raised as an issue by many interviewees…’

Some of the results sections only include one sentence – this makes the manuscript harder to read and suggests that there may not be enough data in some of the categories to describe them in detail. Can you merge some of these sub-headings, or develop them more so that they are not just one phrase or sentence?

This sentence is not clear to me: 'They referred to an IRB decision that forced to exclude those trial participants that initially gave consent but loose the decision making competence over the course of the trial due to disease progression'

Are the potential response strategies the same as the mitigation measures mentioned in the abstract? I would make this clearer and ensure that the language used to describe recommendations/response strategies/mitigation factors is consistent.

Discussion

More direct discussion of the results would be beneficial here, including an explanation of how they link to the informed consent process. I would summarise the results first, reflect upon them, situate your findings within the wider context of why informed consent and trial prioritisation is important and then present your recommendations, whilst also drawing on other relevant literature.

The example of research in LMICs feels a little out of place as the rest of the manuscript is focussing on Germany. The point about varying contexts could either be developed in more detail, or the reference removed.

Nice reflection on the limitations of the study – thank you.

Minor points

A thorough grammar check is required - there are a lot of small errors throughout the manuscript eg first sentence of the introduction should read ‘parts’ not ‘part’ and ‘too little patients’ should read ‘too few’

Please also check the use of tense throughout, as it is not consistent.

Can you explain what a ‘member of public funder is’? Does this mean a ‘member of a public funding body’ or ‘public donor’?

Whilst the meaning of quotations should not be changed or edited, it would be advisable to edit them to make them easier to read.

The use of the word ‘incapable’ would be better amended to ‘unable’.

The age ranges in the table need to be modified - 45 appears in both 35-45 and 45-50

Principal investigator rather than ‘principle’ investigator

I am not familiar with the context thus cannot be sure, but would it be possible for the ‘non-PI investigator’ or the ‘member of public funder’ to be identified? There is only one in each category, so I wonder if readers familiar with the context and the networks of the authors would be able to identify them.

6. PLOS authors have the option to publish the peer review history of their article (what does this mean?). If published, this will include your full peer review and any attached files.

Reviewer #1: No

Reviewer #2: No

---

## [Editor Report · Decision Letter 1]

20 Feb 2024

PONE-D-23-01691R1Informed consent and trial prioritization for clinical studies during the COVID-19 pandemic. Stakeholder experiences and viewpoints.PLOS ONE

Dear Dr. Strech,

Thank you for submitting your manuscript to PLOS ONE. After careful consideration, we feel that it has merit but does not fully meet PLOS ONE’s publication criteria as it currently stands. Therefore, we invite you to submit a revised version of the manuscript that addresses the points raised during the review process.

We look forward to receiving your revised manuscript.

Kind regards,

Augustina Koduah

Academic Editor

PLOS ONE

Journal Requirements:

Additional Editor Comments:

Reviewers comments are adequately addressed.

Consider moving the content under heading ‘demographics’ to the methods section.

-Add – ‘We conducted 21 semi-structured in-depth interviews with 21 participants to reach thematic saturation. Altogether, we contacted and invited 51 stakeholders. Our sample included physicians working in clinical research (principal investigators and non-principal investigators), heads of clinical departments that were responsible for the enrollment of patients with COVID-19 in clinical studies, representatives from clinical study centers, members of research ethics committees, and a member of a public funding body. For complete demographics see table 1.’ To data collection

-Add- ‘For the first topic informed consent we identified three main themes: consent challenges, impact of consent challenges on clinical research, and potential strategies for consent challenges. For the second topic prioritization of clinical studies, we identified two main themes: perceived benefit of prioritization, potential strategies for prioritization. All main themes were further specified with first and second level subthemes (see table 2 for an overview). In the following we give a narrative overview of the identified main and subthemes together with quotes for selected subthemes. The corresponding original quotes from the interviews for all subthemes can be found in supplementary table S2.’ To data analysis

Minor:

‘…consenting procedureswith every COVID-19 patient’ under the Discussion section. Kindly revise to – ‘…consenting procedures with every COVID-19 patient’

---

## [Author Response · Author response to Decision Letter 1]

20 Mar 2024

Additional Editor Comments:

Reviewers comments are adequately addressed.

Consider moving the content under heading ‘demographics’ to the methods section.

-Add – ‘We conducted 21 semi-structured in-depth interviews with 21 participants to reach thematic saturation. Altogether, we contacted and invited 51 stakeholders. Our sample included physicians working in clinical research (principal investigators and non-principal investigators), heads of clinical departments that were responsible for the enrollment of patients with COVID-19 in clinical studies, representatives from clinical study centers, members of research ethics committees, and a member of a public funding body. For complete demographics see table 1.’ To data collection

Authors’ response: We moved the content accordingly. 

-Add- ‘For the first topic informed consent we identified three main themes: consent challenges, impact of consent challenges on clinical research, and potential strategies for consent challenges. For the second topic prioritization of clinical studies, we identified two main themes: perceived benefit of prioritization, potential strategies for prioritization. All main themes were further specified with first and second level subthemes (see table 2 for an overview). In the following we give a narrative overview of the identified main and subthemes together with quotes for selected subthemes. The corresponding original quotes from the interviews for all subthemes can be found in supplementary table S2.’ To data analysis

Authors’ response: We moved the content accordingly but kept the last two sentences in the results section as they directly introduce the following results sections.

Minor:

‘…consenting procedureswith every COVID-19 patient’ under the Discussion section. Kindly revise to – ‘…consenting procedures with every COVID-19 patient’

Authors’ response: We changed the text accordingly.

---

## [Editor Report · Decision Letter 2]

8 Apr 2024

Informed consent and trial prioritization for clinical studies during the COVID-19 pandemic. Stakeholder experiences and viewpoints.

PONE-D-23-01691R2

Dear Dr. Strech,

We’re pleased to inform you that your manuscript has been judged scientifically suitable for publication and will be formally accepted for publication once it meets all outstanding technical requirements.

Kind regards,

Augustina Koduah

Academic Editor

PLOS ONE